# Validation of a Method for Anacardic Acid Quantification in Cashew Peduncles via High-Performance Liquid Chromatography Coupled to a Diode-Array Detector

**DOI:** 10.3390/foods12142759

**Published:** 2023-07-20

**Authors:** Francisco Oiram Filho, Morgana Pereira Mitri, Guilherme Julião Zocolo, Kirley Marques Canuto, Edy Sousa de Brito

**Affiliations:** 1Department of Chemical Engineering, Federal University of Ceará, Fortaleza 60440-900, CE, Brazil; oiramfilho@yahoo.com.br (F.O.F.);; 2Embrapa Agroindústria Tropical, Pici, Fortaleza 60511-110, CE, Brazil; 3Embrapa Alimentos e Territórios, Maceió 60020-050, AL, Brazil

**Keywords:** anacardic acid, *Anacardium occidentale*, HPLC, liquid chromatography, validation

## Abstract

The cashew peduncle has a high nutritional value and contains a wide variety of phenolic compounds. Among these, anacardic acids (AnAc) are biologically active components; however, they influence the cashew juice flavor and, consequently, its acceptance. This study validates a high-performance liquid chromatography method for quantifying the AnAc present in cashew peduncles, using a C_18_ reverse-phase column and a diode-array detector. The calibration curve obtained showed satisfactory precision for intraday (CV = 0.20%) and interday (CV = 0.29%) quantification, linearity (y = 2333.5x + 2956.2; r^2^ = 0.9979), repeatability with respect to retention time (CV = 0.45%) and area (CV = 0.30%), and selectivity, and possessed detection and quantification limits of 0.18 and 0.85 µg·mL^−1^, respectively. Different cashew clones containing AnAc were extracted and analyzed using the proposed method. A recovery of >90% was achieved using two sequential extractions. The total AnAc content ranged from 128.35 to 217.00 mg·100 g^−1^ in peduncle samples obtained from five different cashew clones.

## 1. Introduction

Although cashew cultivation focuses mainly on nuts, the cashew peduncle plays an essential role in the production chain, representing a global production of approximately 1.3 million tons in 2019 [1]. The cashew peduncle has been gaining relevance because of its nutritional composition (rich in carotenoids, vitamin C, and phenolic compounds) [2], which has generated interest in the food and pharmacological industries. However, its consumption in natura remains low due to its high astringency [3] and perishability [4]; therefore, cashew peduncle has been commercialized mainly in the form of industrialized products such as juices, sweets, and jellies.

Anacardic acids (AnAc) are salicylic acid-derived compounds present in nearly all parts of the cashew tree, including the cashew peduncle. Over the past few years, many studies have reported the biological activities of anacardic acids, such as their enzyme inhibitory (e.g., anticholinesterase) [5], bactericidal [6], antitumor [7], antioxidative [8], and antiprotozoal [9] activities. Owing to these activities, AnAc have gained significant importance in several fields.

Structurally, AnAc are classified as phenolic lipids because they have a long carbon chain with structural variations. The number of carbon atoms present in this chain can vary between 2 and 17 [10,11], with the number of unsaturated C=C bonds ranging between 0 and 3 [11,12]. However, the structural differences between AnAc can give them different biological activities [13] and sensory properties [14].

Astringency is an important sensory aspect of the cashew peduncle. It generates multiple sensations in the mouth, such as bitterness, roughness, and tightness in the cheek muscles when chewing the fruit [15,16]. Previous studies have related the tannins present in the cashew peduncle to this astringency. However, recent studies have proven the influence of AnAc on the astringency of the cashew peduncle [14,17]. Some cultivars exhibit greater astringency than others, which may be related to the AnAc content in the peduncle; therefore, fruits having a higher AnAc content may be more astringent, directly impacting their market acceptance.

There are some analytical quantification methods with high precision for fruit constituents. The most used are gas and liquid chromatography coupled to sensitive detectors such as mass spectrometry and diode array; other techniques also used to quantify these kind of molecules are capillary electrophoresis and near-infrared spectroscopy [18].

Usually, research with anacardic acids is directed towards their biological effect. The development of AnAc quantification methodologies is generally established for the cashew nut shell liquid, since this is the main source of anacardic acids. However, anacardic acids are consumed in the diet mainly via Anacardiaceae fruits such as cashew and mangoes, as well as pistachio nut. To the best of our knowledge, there is a lack of validated methodologies for quantification of these compounds in cahsew peduncles. Thus, owing to the biological activities related to AnAc and their relative link to the astringency of the cashew peduncle, it is necessary to develop and validate a method for quantifying the AnAc present in cashew peduncles. High-performance liquid chromatography (HPLC) is essential to rapidly and efficiently quantify and classify different types of cashew clones according to their AnAc content.

## 2. Materials and Methods

### 2.1. Chemicals and Standards

HPLC-grade acetonitrile and methanol (purity ≥ 99.9%) were purchased from Lichrosolv Merck (Darmstadt, Germany), glacial acetic acid P.A. (purity ≥ 99.9%) and formic acid P.A. (purity ≥ 96%) were from Sigma-Aldrich (Saint Louis, MO, USA), and ultra-pure water (type 1) was obtained from a Milli-Q system (São Paulo, Brazil). The triene AnAc standard (15:3) used to obtain the analytical curve was obtained by the method described previously [11].

### 2.2. Plant Material

Embrapa Agroindustry Tropical kindly provided the cashew peduncles. Five samples from different cashew clones (CCP 09, CCP 76, BRS 265, BRS 275, and Embrapa 51) were analyzed. The cashew clone characteristics can be retrieved at https://www.embrapa.br/busca-de-publicacoes/-/publicacao/1125932/clones-de-cajueiro (accessed on 20 January 2023).

### 2.3. Chromatographic System

The chromatographic system consisted of a Shimadzu LC-20AB Prominence chromatograph coupled to a Shimadzu SPD-M20A Prominence diode array detector and a Shimadzu SIL-20AC Prominence autosampler (Kyoto, Kyoto, Japan). The data were processed using Shimadzu LC Solution software v5.82. The analytical conditions used were a reverse-phase C18 chromatographic column Shim-pack CLC–ODS (M) (150 × 4.6 mm × 5 μm, Shimadzu, Kyoto, Japan). The mobile phase used was water (Solvent A) and acetonitrile (Solvent B) in a ratio of 20:80, both acidified with acetic acid (1%), running in isocratic mode. The analysis time was 30 min, with a flow of 1.5 mL min^−1^, at 25 °C, with an injection volume of 20 μL. The chromatograms were monitored at a wavelength of 280 nm and the UV spectra were recorded from 200 to 400 nm; these conditions were reported previously [19].

### 2.4. Extraction of AnAc from Cashew Peduncles

The peduncles were sliced and freeze-dried (LioBras K105, São Carlos, SP, Brazil) for 96 h for the complete removal of water. Subsequently, the samples were ground in an electric grinder (Cadence MDR302-127, Navegantes, SC, Brazil) to obtain a fine powder.

For each extraction, 0.5 g of powder was weighed, and 9 mL of methanol was added. Five sequential extractions were performed on the cashew peduncle powder to extract the AnAc from the samples completely. Each extraction was performed for 20 min using an Ultrasonic Cleaner 1400 (THORTON/UNIQUE) at room temperature, after which the mixture was centrifuged (Kindly KC5, São Paulo, SP, Brazil) for 10 min at 2950× *g*. The supernatant obtained from each extraction was filtered through filter paper and evaporated by vacuum distillation (Buchi V-850, Valinhos, SP, Brazil). Each supernatant was analyzed separately to obtain the percentage extraction for every stage to ensure the complete extraction of the target analytes.

### 2.5. Structural Confirmation of AnAc by UPLC–QTOF-MS^E^

To unequivocally identify and confirm the three different AnAc (15:3, 15:2, and 15:1) in the sample extracts, an analysis was performed according to the methodology previously cited [19] with adjustments. An Acquity UPLC system (Waters, Milford, MA, USA) coupled to a quadrupole time-of-flight (QToF) mass spectrometer (Waters, Milford, MA, USA) was used under the following conditions: chromatographic column Acquity BEH C18 (150 × 2.1 mm^2^, 1.7 μm; Waters, Milford, MA, USA), operated at 40 °C. The eluent system employed was a combination of A (0.1% formic acid in water) and B (0.1% formic acid in acetonitrile) at a flow rate of 0.3 mL min^−1^. The gradient varied linearly from 5 to 95% B (*v*/*v*) over 30 min. The sample injection volume was 5 μL. Mass spectra were obtained in the negative-ion mode over a mass range between 50 and 1180 Da. The spectrometer was operated with MS^E^ centroid programming using a cone voltage of 40 V. The drying gas pressure was 35 psi at 370 °C, while the nebulizer gas pressure was 40 psi. A capillary voltage of 3500 V and a 600 V spray shield voltage were used. The structural identification of the anacardic acids was carried out using the molecular formulae and *m/z* values obtained from the high-resolution mass spectra. The chromatographic peaks had higher intensities, as determined using MassLynx software v4.1 (Waters Corporation). The structural analyses for molecules of AnAc were performed by comparing the fragmentation patterns from the MS/MS data [20]. The data were compared to those from previous reports [12,21].

### 2.6. Validation Method

The validation method was established according to the rules adopted in the International Conference on Harmonization (ICH) [22], considering the quality, consistency, and reliability of the obtained results. The parameters verified were the selectivity, linearity, precision (intraday and interday), system suitability, accuracy, recovery, limit of detection (LOD), and limit of quantification (LOQ). A linearity plot was prepared at eight different concentrations (1, 5, 10, 20, 40, 60, 80, and 100 µg·mL^−1^ or 0.003, 0.014, 0.029, 0.058, 0.116, 0.173, 0.231 and 0.289 µmol·mL^−1^) of the external AnAc triene standard (15:3). Furthermore, five samples of different clones of cashew apples produced in the experimental fields of the Embrapa Tropical Agroindustry were quantified to demonstrate the applicability of the validated method. The extracts obtained from the sequential extractions were solubilized in 3 mL of methanol filtered on a 45 µm PTFE disc filter for the quantification of AnAc by HPLC–DAD. Each extract from the five extraction steps was analyzed separately.

### 2.7. Selectivity

Selectivity is defined as the capacity of a method to unequivocally identify an analyte present in a mixture or matrix without interference from other compounds [22,23]. To evaluate the purity of the chromatographic peaks, the three-point purity of each peak in the UV spectrum was compared to the UV spectra of the external standard, samples, and blank using Shimadzu LC Solution software. The peaks whose UV spectra (220–400 nm) had a similarity higher than 95% were considered pure.

### 2.8. Linearity

To verify the linearity of the method, an analytical curve was constructed using an external standard of the AnAc triene (15:3) solubilized in methanol at eight different concentrations (1, 5, 10, 20, 40, 60, 80, and 100 µg·mL^−1^), and all tests were performed in triplicate. We applied Student’s *t* test and F test with a 95% confidence level to ensure that the data obtained from the calibration curve and linear regression were highly reliable. The calculations were performed using the following equations:(1)$$S^{2}y=\frac{\sum {(d}^{2}i)}{n-2}$$
(2)$$S^{2}a=\frac{S^{2}y\cdot n}{D}$$
(3)$$S^{2}b=\frac{S^{2}y\cdot \sum \left(x^{2}i\right)}{D}$$
where *Sy* is the standard deviation on the *y*-axis, *Sa* and *Sb* are the standard deviations of the slope (*a*) and linearity (*b*), respectively, *xi* are the individual values of *x*, *n* is the total number of points on the curve, *di* is the vertical deviation of each point, and *D* is the determinant given as
$$\left|\begin{array}{cc} \sum x^{2}i & \sum xi\\ xi & n \end{array}\right|$$

From the standard deviations calculated in Equations (1)–(3), the values of *t_calc_*_(*a*)_, *t_calc_*_(*b*)_, and *F_calc_* were calculated using the following equations:(4)$$t_{calc(a)}=\frac{\left|1-a\right|}{S_{a}}$$
(5)$$t_{calc(b)}=\frac{\left|b\right|}{S_{b}}$$
(6)$$F_{calc}=\frac{\frac{S_{b}}{1}}{\frac{\sum d_{i}}{3}}$$

The values calculated by the equations above were compared to tabulated values (*t_tab_* and *F_tab_*) acquired by the tables from Student’s *t* test and F test. When *t_calc_* > *t_tab_*, the parameter is significant at a 95% confidence level and must remain on the curve. When *t_calc_* < *t_tab_*, the parameter is insignificant and can be excluded. For the F test, if *F_calc_* > *F_tab_*, the parameters are significant at the 95% confidence level for the linear regression. If *F_calc_* < *F_tab_*, regardless of the value of the coefficient of determination (R^2^), the parameters are insignificant, indicating that there is no linear relation between the *x*- and *y*-axes [19].

### 2.9. Precision

The method’s precision was determined by assessing the reproducibility of the eight different concentrations of the external standard (1, 5, 10, 20, 40, 60, 80, and 100 µg·mL^−1^). The reproducibility of the triplicates at each point on the analytical curve was evaluated on the same day (intraday) and subsequently on three different days (interday). The precision was calculated from the standard deviations and coefficients of variation as follows:(7)$$S=\sqrt{\frac{\sum_{i=1}^{n}{\left(x_{i}-\bar{x}\right)}^{2}}{n-1}}$$
(8)$$Cv=\frac{s}{\bar{x}}\cdot 100$$
where *S* is the standard deviation, *n* is the number of measurements, *x_i_* is the value of each individual measure, and $\bar{x}$ is the mean.

### 2.10. Accuracy and Recovery

Some protocols may be applied to different analytes to evaluate the accuracy and recovery of the chromatographic method. One of these protocols is doping the sample using the target analyte at a known concentration [22,24]. However, some samples cannot be doped to ensure a recovery that is close to adequate, as stipulated by the validation guidelines. However, exhaustive extraction ensures that the target analyte present on the sample is extracted completely, thereby enabling the evaluation of the analyte recovery by the method developed [25].

The spectra of each analyte from the five sequential extractions were used to support the recovery calculations. Therefore, it was possible to calculate the recovery percentage of each of the five extractions for the three different AnAc (triene, diene, and monoene) using the following equation:(9)$$R\left(\%\right)=\frac{A_{x}}{\sum A_{n}}\;\cdot \;100$$
where *A_x_* is the chromatographic peak area for each analyte in each of the five extractions and *∑A_n_* is the total summatory area under the chromatographic peaks of the five sequential extracts for each analyte.

### 2.11. System Suitability Test

The system suitability was determined from the repeatability of an intermediate point on the calibration curve (40 µg·mL^−1^) following 10 consecutive injections under the same operating conditions. The standard deviation and coefficient of variation were calculated from the retention times and areas obtained [22].

### 2.12. Limit of Detection and Quantification

The limit of detection (LOD) represents the lowest concentration at which a substance can be detected, i.e., the analyte can be differentiated from noise. The limit of quantification (LOQ) represents the lowest concentration at which a substance can be quantified with accuracy and fidelity.

The LOD and LOQ parameters were based on visual evaluation, as described in the international standard for validation [22].

## 3. Results

### 3.1. Method Development

The chromatographic separation was the same as described in the literature [19]. The tests carried out previously showed that the optimal chromatographic conditions for separating the AnAc present in the sample were as follows: a reverse-phase C_18_ column (150 mm × 4.6 mm × 5 µm) was used. The mobile phase was supplied at a flow rate of 1.5 mL·min^−1^ in an isocratic manner with 80% acetonitrile. The mobile phase was acidified using acetic acid to enhance the resolution of the chromatographic peaks.

The peak for each AnAc was identified using its retention time and UV spectrum. Figure 1A shows the chromatographic profile of the external standard of triene AnAc (15:3). Figure 1B shows the chromatographic profile of the cashew peduncle, in which triene (15:3), diene (15:2), and monoene (15:1) AnAc were identified.

### 3.2. Method Validation

The method was validated based on the principles and parameters provided by the International Conference on Harmonization (ICH) [22]. The results obtained for selectivity, linearity, precision, system suitability, limits of quantification, detection, and accuracy or recovery were based on the standards required for each parameter mentioned.

### 3.3. Selectivity and Linearity

Figure 2 shows the selectivity results, which were evaluated based on the UV spectrum of the external standard of triene AnAc (15:3). The purities calculated from the three peaks were 99.17%, 98.19%, and 99.40% for (15:3), (15:2), and (15:1) AnAc, respectively.

The calibration curve was constructed from eight concentrations (1, 5, 10, 20, 40, 60, 80, and 100 µg·mL^−1^) of triene AnAc (15:3). Table 1 lists the linearity of the curve, which was confirmed by the value of the coefficient of determination of the analytical curve equation (y = 2333.5x + 2956.2; R^2^ = 0.998), which is in agreement with those found in the literature and validation guides [24,26]. To verify the reliability of the calibration data of the curve and the linear regression, Student’s *t* test and the F test were used. The angular and linear coefficients of the calibration curve were significant; therefore, both terms play important roles in the quantification of the compounds and must remain in the equation. The F test showed that there is a linear relation between the *x*- and *y*-axes because *F_cal_* was greater than *F_tab_*.

### 3.4. Precision and System Suitability Tests

Table 1 lists the precisions of the intraday and interday analyses, and the values obtained for the determination coefficients and the coefficients of variation were R^2^ = 0.999, CV = 0.20% and R^2^ = 0.998, CV = 0.29% for the intraday and interday analyses, respectively. The coefficients of variation calculated for the system suitability test, performed at the median of the analytical curve for retention time and peak area, were 0.45% and 0.30%, respectively. All the values related to system precision and suitability were in accordance with the parameters established in the established validation guidelines, with determination coefficients R^2^ ≥ 0.99 and relative standard deviations (RSD) of approximately 1.0% [22,24,25,27].

### 3.5. LOD and LOQ

Table 1 lists the LOD and the LOQ as 0.18 and 0.83 µg·mL^−1^, respectively. These values were obtained visually in accordance with international validation guides and showed that the method has a high sensitivity for determining AnAc.

### 3.6. Accuracy or Recovery

The accuracy of the method was based on the exhaustive extraction of the target analyte, and this technique may be applied when the sample cannot be doped efficiently to ensure the correct evaluation of the accuracy. The results presented in Table 2 show the quantification of AnAc in each of the five extracts. The results from all the analyzed samples obtained from the first and second sequential extractions were viable for quantifying the compounds. However, from the third sequential extraction onward—the samples cashew peduncle CCP09, BRS275, and Embrapa51—showed exhaustion of the target compounds (triene and diene AnAc) to below the limit of quantification, which proves the viability of the method using two sequential extractions. The total values of amount from each sample of AnAc triene, diene, and monoene were 24.40, 18.47 and 127.72 mg·100 g^−1^ respectively, for CCP09; 31.16, 56.87, and 111.96 mg·100 g^−1^, respectively, for CCP76; 44.76, 37.38, and 134.86 mg·100 g^−1^, respectively, for BRS265; 29.83, 19.15, and 93.63 mg·100 g^−1^, respectively, for BRS275; and 19.62, 13.71, and 95.02 mg·100 g^−1^, respectively, for Embrapa 51. Table 3 shows the recovery from each of the five extractions for all samples analyzed. The percentages of recovery from the two first extractions were as follows: CCP09—96.77, 96.15, and 96.24% of triene, diene, and monoene, respectively; CCP76—92.22, 91.90, and 91.83%, respectively; BRS265—96.59, 96.73, and 96.38%, respectively; BRS275—95.89, 95.60, and 96.31%, respectively; and Embrapa 51—95.91, 95.80, and 95.75%, respectively.

### 3.7. Structural Confirmation of AnAc and Quantification of a Real Sample

The identification of the AnAc was confirmed using mass spectrometry, considering their exact masses and ionization fragments. Three AnAc were identified (Figure 3) in the samples; triene (15:3), diene (15:2), and monoene (15:1) AnAc presented deprotonated ions [M−H]^−^ at *m*/*z* 341.2079, 343.2228, and 345.2405, respectively, in addition to their respective ion fragments at *m*/*z* 297.2187, 299.2332, and 301.2505 (Figure 4). These molecular masses and their fragmentation patterns have been reported in previous studies [2,19,28].

## 4. Discussion

The HPLC–DAD analyses might show some limitations such as problems with selectivity because more than one compound can have the same absorption UV spectra or similar retention behavior, leading to an equivocal identification and quantification [29]. Sensitivity is another limitation which can occur due to a weak absorption UV or lower concentrations levels of analyte [30]. Even with these limitations, HPLC–DAD is still a tool widely used for compound quantification [31,32,33], since good sample preparation minimizes these problems. Furthermore, the HPLC–DAD systems can allow for easy method development when compared to other chromatography systems.

For method development, some characteristics of compounds must be analyzed in the case of AnAc: the long carbon chains and variation in unsaturation along the chains render AnAc nonpolar. Therefore, chromatographic columns that have a greater amount of carbon attached to the stationary phase provide a better chromatographic resolution [34]. Thus, the C_18_ column, which contains 18 carbon atoms chemically attached to silicon in its stationary phase, had a higher chromatographic resolution than the C_8_ column, containing 8 attached carbon atoms. Acetonitrile exhibited greater efficiency in separating the three AnAc because of its higher elution strength compared to methanol [35]. Parameters, such as the percentage of organic solvent and elution flow, were optimized based on the principles of lower concentration and shorter run time to separate the AnAc. The pH of the mobile phase was set at 3.0 to improve the peak resolution by preventing the formation of tail effects due to the inactivation of residual silanols present in the stationary phase [36].

The method provides important data to the literature because there is no other published validated method that is capable of quantifying AnAc in cashew apple peduncles. The results obtained for the evaluated parameters showed that the methodology developed using HPLC–DAD could be used to monitor and quantify the anacardic acids present in cashew peduncles. The validated method is efficient because it presented detection levels appropriate to the concentrations of the anacardic acids of interest, providing reliable and accurate data rapidly, which is suitable for laboratory needs [35].

The content of AnAc in five different analyzed samples (Table 2) had similar profiles when compared to the results from the literature, wherein the major AnAc were monoene (15:1), and the minor AnAc were triene (15:3). One of the works evaluating the content of alkylphenols from cashew peduncle reported values of 22.00, 32.00, and 56.00 mg·100 g^−1^ for triene, diene, and monoene AnAc, respectively [12]. However, the content of triene and diene AnAc reported in the literature vary between 19.62 and 44.76 mg·100 g^−1^ for triene and between 13.71 and 56.87 mg·100 g^−1^ for diene. In contrast, the content of the monoene AnAc ranged between 93.63 and 134.86 mg·100 g^−1^ for samples BRS275 and BRS265, respectively. This variation in AnAc content is due to the genetic variation in the clones analyzed.

The recovery percentage presented In Table 3 shows that the first extraction obtained values between 70.25% and 85.35% for samples CCP76 and CCP09, respectively. Values between 70% and 130% are required by the international validation standards used [22,37,38]. However, the second extraction represents a significant portion of the recovery of each analyte. The recovery values for the second extraction varied between 11.19% and 21.58% for samples BRS275 and CCP76, respectively. Thus, the method showed an excellent recovery of analytes, varying between 91.84 and 96.77% for samples CCP76 and CCP09, respectively, using two sequential extractions. These percentages are similar to those obtained by other validation methods [39,40,41,42]. All the results presented in Table 3 were obtained from Equation (9).

## 5. Conclusions

The chromatographic method developed was fast, precise, accurate, and capable of efficiently quantifying the three main AnAc present in the cashew peduncle using HPLC. The method also presents itself as an alternative for establishing quality-control protocols, which can be applied in industrial material-sealing processes for the food industry and in advanced phenotyping protocols for genetic breeding programs. Therefore, the development of this method is of great importance in cashew production to assist in the classification of peduncles and the choice of cashew clones according to the desired AnAc content.

## Figures and Tables

**Figure 1 foods-12-02759-f001:**
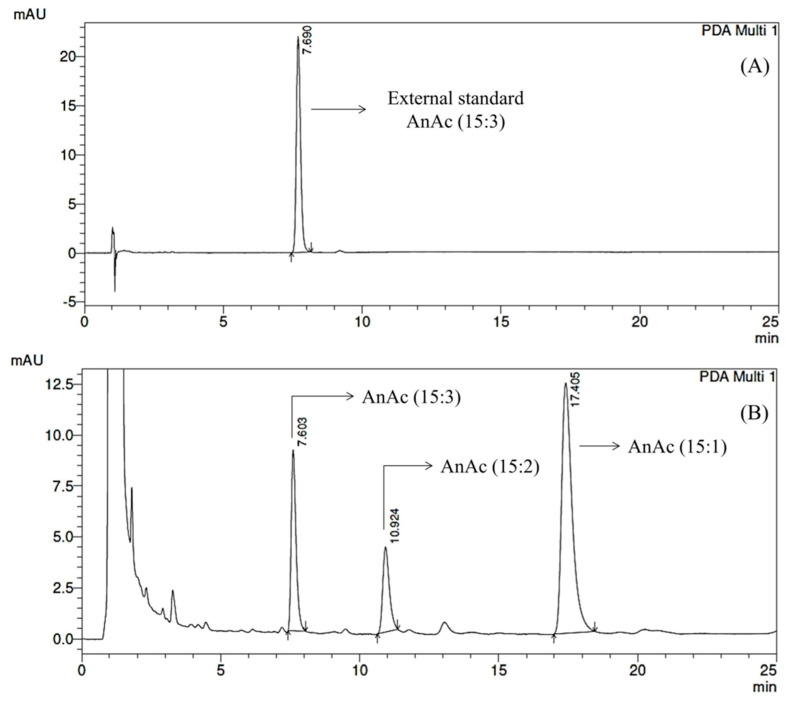
(**A**) Chromatographic profile of the external standard of triene AnAc (15:3). (**B**) Chromatographic profile of AnAc present in the cashew peduncle from sample BRS275, analyzed by HPLC–DAD and monitored at 280 nm.

**Figure 2 foods-12-02759-f002:**
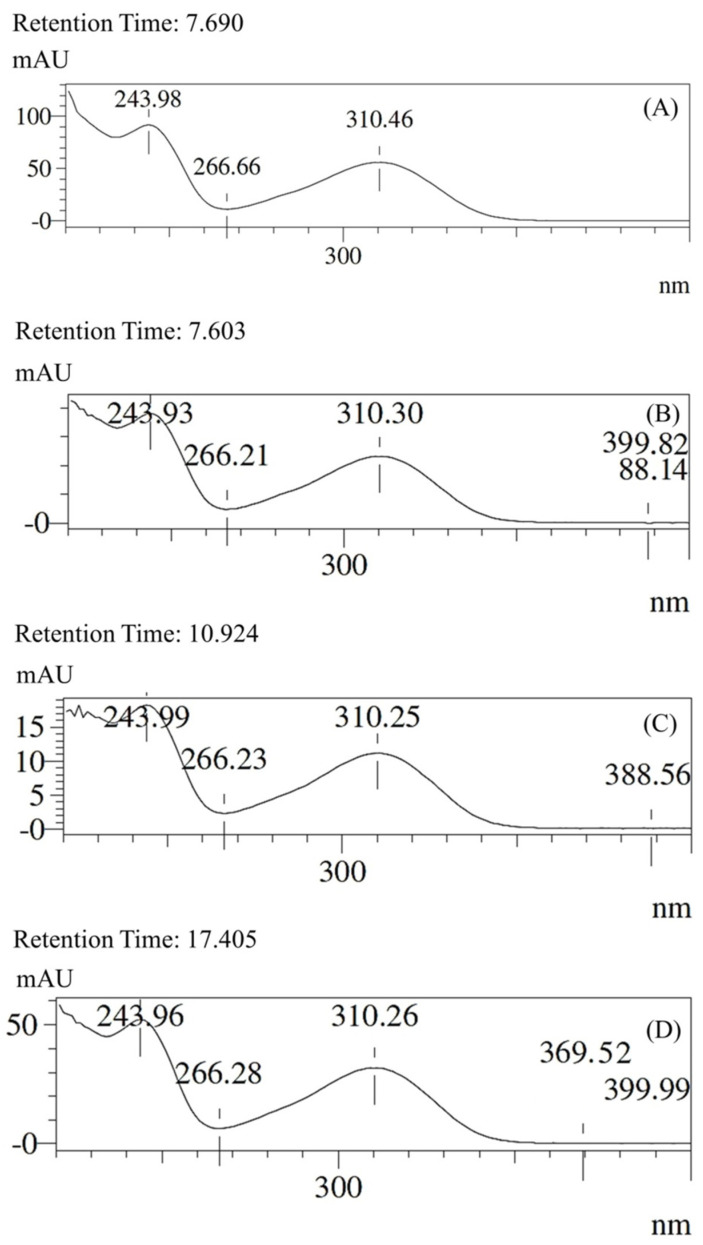
UV spectra of the (**A**) external standard triene AnAc (15:3), (**B**) triene AnAc (15:3), (**C**) diene AnAc (15:2), and (**D**) monoene AnAc (15:1). The UV spectra (**B**–**D**) are from cashew peduncle sample BRS275. All UV spectra were recorded between 220 and 400 nm.

**Figure 3 foods-12-02759-f003:**
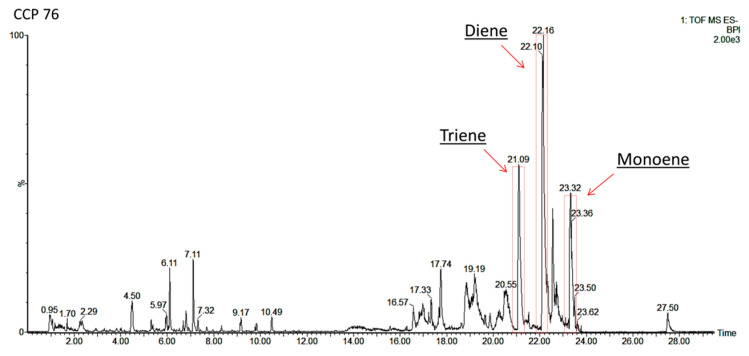
Chromatographic profile of AnAc (15:3), diene AnAc (15:2), and monoene AnAc (15:1) from sample cashew peduncle CCP 76, analyzed by UPLC–QTOF-MS^E^.

**Figure 4 foods-12-02759-f004:**
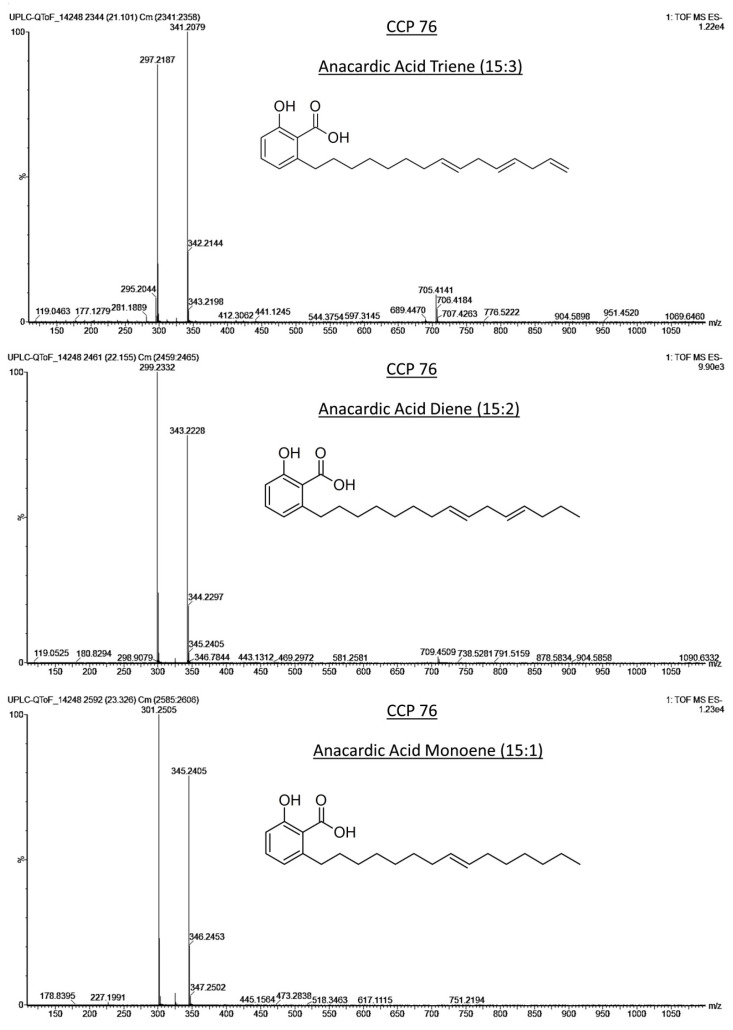
Mass spectra of anacardic acids, triene (15:3), diene (15:2), and monoene (15:1) obtained from the chromatographic profile of sample CCP 76, analyzed by UPLC–QTOF-MS^E^.

**Table 1 foods-12-02759-t001:** Statistical results for the linear regression of the calibration curve of the external standard, Student’s *t* test, and F test, at 95% confidence levels.

Analytical Curve	y = 2333.5x + 2956.2		R^2^ = 0.998		
Angular Coefficient	A	*S_a_*	*t_cal_*	*t_tab_*	*t* test
	2333.5	15.79	147.63	2.009	Significant
Linear Coefficient	B	*S_b_*	*t_cal_*	*t_tab_*	*t* test
	2956.2	830.91	3.56	2.009	Significant
*F_calc_*	*F_tab_*		*F* Test		
21,812.56	1.576		Significant		
	R^2^		CV (%)		
Intraday (n = 6)	0.9993		0.20		
Interday (n = 9)	0.9979		0.29		
		Retention time (RSD)		Area (RSD)	
System suitability test		0.45		0.30	
		Concentration (µg·mL^−1^)			
LOD		0.18			
LOQ		0.85			

**Table 2 foods-12-02759-t002:** Quantification of triene (15:3), diene (15:2), and monoene (15:1) AnAc from 5 sequential extractions and total amount extracted, performed on cashew peduncles obtained from different cashew clones, both expressed as mg·100 g^−1^.

Sample	AnAc	Extraction (mg·100 g^−1^)					Total Amount (mg·100 g^−1^)
		1	2	3	4	5	
CCP 09	15:3	22.07 ± 0.13	2.32 ± 0.01	<LOQ	<LOQ	<LOQ	24.40 ± 0.14
	15:2	16.66 ± 0.02	1.81 ± 0.04	<LOQ	<LOQ	<LOQ	18.47 ± 0.02
	15:1	109.88 ± 0.42	14.63 ± 0.14	2.39 ± 0.09	0.81 ± 0.02	<LOQ	127.72 ± 0.48
CCP 76	15:3	23.16 ± 0.06	6.34 ± 0.07	1.65 ± 0.01	<LOQ	<LOQ	31.16 ± 0.11
	15:2	41.22 ± 0.41	12.01 ± 0.09	3.62 ± 0.05	<LOQ	<LOQ	56.87 ± 0.34
	15:1	79.98 ± 0.28	24.07 ± 0.33	7.74 ± 0.31	0.16 ± 0.04	<LOQ	111.96 ± 0.29
BRS 265	15:3	39.88 ± 0.31	4.64 ± 0.06	0.23 ± 0.02	<LOQ	<LOQ	44.76 ± 0.32
	15:2	33.29 ± 0.19	3.99 ± 0.03	0.10 ± 0.01	<LOQ	<LOQ	37.38 ± 0.22
	15:1	115.26 ± 0.65	16.14 ± 0.04	2.33 ± 0.10	1.12 ± 0.89	<LOQ	134.86 ± 1.14
BRS 275	15:3	26.88 ± 0.25	2.95 ± 0.06	<LOQ	<LOQ	<LOQ	29.83 ± 0.29
	15:2	17.51 ± 0.11	1.64 ± 0.10	<LOQ	<LOQ	<LOQ	19.15 ± 0.04
	15:1	81.26 ± 2.04	10.45 ± 0.19	1.42 ± 0.15	0.49 ± 0.18	<LOQ	93.63 ± 2.01
Embrapa 51	15:3	16.56 ± 0.08	3.05 ± 0.01	<LOQ	<LOQ	<LOQ	19.62 ± 0.07
	15:2	11.66 ± 0,05	2.05 ± 0.03	<LOQ	<LOQ	<LOQ	13.71 ± 0.06
	15:1	75.95 ± 0.82	16.56 ± 0.58	2.09 ± 0.03	0.40 ± 0.06	<LOQ	95.02 ± 1.10

<LOQ: Values below the limit of quantification.

**Table 3 foods-12-02759-t003:** Recovery percentages of triene (15:3), diene (15:2), and monoene (15:1) AnAc from five sequential extractions performed on different cashew peduncles.

Sample	AnAc	Recovery (%)				
		1	2	3	4	5
CCP09	15:3	85.35 ± 0.49	11.42 ± 0.05	2.18 ± 0.16	0.93 ± 0.05	0.11 ± 0.05
	15:2	83.88 ± 0.13	12.27 ± 0.22	2.56 ± 0.06	1.14 ± 0.06	0.13 ± 0.06
	15:1	84.51 ± 0.32	11.73 ± 0.11	2.18 ± 0.07	1.17 ± 0.02	0.19 ± 0.06
CCP76	15:3	71.17 ± 0.19	21.05 ± 0.22	7.06 ± 0.04	0.66 ± 0.05	0.03 ± 0.02
	15:2	70.48 ± 0.70	21.42 ± 0.15	7.30 ± 0.09	0.78 ± 0.10	0.02 ± 0.02
	15:1	70.25 ± 0.25	21.58 ± 0.29	7.36 ± 0.27	0.77 ± 0.03	0.02 ± 0.01
BRS265	15:3	85.33 ± 0.66	11.26 ± 0.13	2.00 ± 0.05	1.33 ± 0.77	0.05 ± 0.04
	15:2	84.97 ± 0.48	11.76 ± 0.08	2.03 ± 0.03	1.16 ± 0.74	0.06 ± 0.03
	15:1	84.15 ± 0.47	12.23 ± 0.03	2.21 ± 0.07	1.33 ± 0.65	0.06 ± 0.01
BRS275	15:3	84.56 ± 0.79	11.33 ± 0.20	2.34 ± 0.10	1.54 ± 0.05	0.22 ± 0.05
	15:2	84.41 ± 0.51	11.19 ± 0.47	1.90 ± 0.23	1.31 ± 0.07	0.18 ± 0.07
	15:1	84.73 ± 2.11	11.58 ± 0.19	2.24 ± 0.16	1.29 ± 0.19	0.15 ± 0.04
Embrapa51	15:3	78.67 ± 0.36	17.24 ± 0.05	2.74 ± 0.03	1.17 ± 0.18	0.17 ± 0.02
	15:2	78.24 ± 0.30	17.56 ± 0.18	2.73 ± 0.14	1.17 ± 0.20	0.28 ± 0.32
	15:1	78.13 ± 0.84	17.62 ± 0.59	2.88 ± 0.03	1.15 ± 0.06	0.20 ± 0.05

## Data Availability

Data is contained within the article.

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
