# Peer review of "Validation of a Method for Anacardic Acid Quantification in Cashew Peduncles via High-Performance Liquid Chromatography Coupled to a Diode-Array Detector"

_foods, 2023, doi:10.3390/foods12142759_

Round 1
Reviewer 1 Report
The present manuscript entitled "Validation of a Method for Anacardic Acid Quantitation in Cashew Peduncles via High-Performance Liquid Chromatography Coupled to a Diode-Array Detector" by Francisco Oiram Filho, Morgana Mitri, Guilherme Julião Zocolo, Kirley Marques Canuto, and Edy Sousa de Brito (foods-2454145) is written correctly and has a good structure; moreover, it has all the necessary parts. The article is interesting from an analytical point of view; therefore, it should interest the reader. My current decision is a minor revision.
1. Units. The dot in the unit should be placed higher.
2. How is the recovery calculation susceptible to interference effects?
3. Introduction. Please add a section on analytical methods used for this type of research.
4. Page 2, line 65. What were the parameters of the water used?
5. Page 2, line 68. “2.1” should be instead of “1.2”. The title is too far left.
6. Section 2.3. Please, provide more detail about analytical conditions shortly.
7. Section 2.4. Ultrasonic extraction. At what temperature was the extraction performed?
8. Page 3, line 132. What can be done in the event of strong interference effects? How would you deal with them? What types of interference effects could occur?
9. Page 4, line 172, and page 7, line 266. The "relationship" is mentioned. This term should be changed to "relation". The relationship tends to be used more broadly to describe the interactions between specific people or smaller groups of people.
10. Page 5, lines 219-220. LOD and LOQ can also be calculated from the calibration chart. This seems to be a better solution than visual designation.
11. Figures. The axes should be named with the unit. Values overlap in one drawing.
12. Use 1, 2, and 3 instead of I, II, and III when numbering the Tables.
13. RSD expressed as a percentage is the coefficient of variation (CV).
14. What are the limitations of the research conducted? You can add a short discussion.
15. Appropriate tools can be used to characterize best the developed method (e.g., AGREE - Analytical GREEnness Metric Approach). This may be of interest to the reader
16. Conclusions. Please clearly highlight the most important advantage of the carried out research.
I hope that the comments presented will help improve the article.
Author Response
We would like to thank the reviewers for their comments. We believe that suggestions contribute to the improvement of our manuscript. Below is a point-by-point response (italic) to each item raised by the reviewers.
Reviewer 1
The present manuscript entitled "Validation of a Method for Anacardic Acid Quantitation in Cashew Peduncles via High-Performance Liquid Chromatography Coupled to a Diode-Array Detector" by Francisco Oiram Filho, Morgana Mitri, Guilherme Julião Zocolo, Kirley Marques Canuto, and Edy Sousa de Brito (foods-2454145) is written correctly and has a good structure; moreover, it has all the necessary parts. The article is interesting from an analytical point of view; therefore, it should interest the reader. My current decision is a minor revision.
- Units. The dot in the unit should be placed higher.
There was probably a misconfiguration in the file. The forwarded file has no such problem.
- How is the recovery calculation susceptible to interference effects?
Matrix effects in HPLC-DAD (High Performance Liquid Chromatography with Diode Array Detector) are caused by the presence of non-target compounds at the same retention time as the analyte of interest. These effects can cause unwanted interactions with the analyte and can affect the results of quantifications in the following ways:
- Signal Interference: Matrix components may have similar absorption spectra to the analyte of interest in the DAD, causing an increase in the measured signal and resulting in an overestimation of the analyte concentration.
- Changes in retention: Matrix components can interact with the stationary phase of the HPLC column, altering the retention time of the analyte of interest. This can result in an underestimation or overestimation of the analyte concentration, depending on the nature of the interactions.
- Suppression or increase in signal: Matrix compounds can change the properties of the solvent, causing a suppression or increase in the analytical signal. If suppression occurs, the analyte concentration may be underestimated. If the increase occurs, it may result in an overestimation.
To overcome matrix effects, strategies such as sample dilution, sample cleanup, or chromatographic separation optimization can be adopted to ensure that the analyte of interest is clearly separated from the matrix components. Additionally, using built-in patterns can also help correct for variations caused by matrix effects.
- Introduction. Please add a section on analytical methods used for this type of research.
Done
- Page 2, line 65. What were the parameters of the water used?
Parameter was Milli Q water type 1
- Page 2, line 68. “2.1” should be instead of “1.2”. The title is too far left.
Done
- Section 2.3. Please, provide more detail about analytical conditions shortly.
Done
- Section 2.4. Ultrasonic extraction. At what temperature was the extraction performed?
Done
- Page 3, line 132. What can be done in the event of strong interference effects? How would you deal with them? What types of interference effects could occur?
Interference effects in chromatographic analysis, such as HPLC-DAD, can be caused by a variety of factors, including co-eluting compounds, mobile phase impurities, or even instrument noise. There are several strategies to handle these interferences:
- Sample Preparation: Improve your sample preparation technique to reduce the presence of interfering substances. This could involve steps like solid-phase extraction (SPE), liquid-liquid extraction, or dilution.
- Optimization of Chromatographic Conditions: You can adjust the mobile phase composition, gradient, or temperature, or even choose a different column to enhance the separation between the analyte and the interfering substances.
- Use of Internal Standards: An internal standard that closely matches the physical and chemical properties of the analyte can be used to correct for any variation in the signal due to interference.
- Use of a Different Detector: If interference is due to overlap in absorption spectra, using a different detector, such as a mass spectrometer, can be helpful. It can provide additional identification and quantification capabilities.
In the context of HPLC-DAD, there are two types of interference effects that could occur:
- Spectral Interference: This occurs when an interfering compound has a similar absorption spectrum to the analyte of interest. It can lead to inaccurate quantification if not appropriately addressed.
- Retention Time Interference: This happens when an interfering compound has a similar retention time to the analyte of interest. It can result in peak merging or peak distortion, which can make quantification difficult.
Dealing with interference effects often requires a careful balance between sample preparation, chromatographic conditions, and detector choice.
- Page 4, line 172, and page 7, line 266. The "relationship" is mentioned. This term should be changed to "relation". The relationship tends to be used more broadly to describe the interactions between specific people or smaller groups of people.
Done
- Page 5, lines 219-220. LOD and LOQ can also be calculated from the calibration chart. This seems to be a better solution than visual designation.
These analytical parameters were defined by visual method according to guidelines used.
- Figures. The axes should be named with the unit. Values overlap in one drawing.
The figures are provided by software according to data from chromatographic peaks. We cannot make changes regarding overlap situation.
- Use 1, 2, and 3 instead of I, II, and III when numbering the Tables.
Done
- RSD expressed as a percentage is the coefficient of variation (CV).
Done
- What are the limitations of the research conducted? You can add a short discussion.
Done
- Appropriate tools can be used to characterize best the developed method (e.g., AGREE - Analytical GREEnness Metric Approach). This may be of interest to the reader.
Absolutely, using appropriate tools to characterize and evaluate the developed method is crucial. One of the modern concepts that is gaining attention in analytical chemistry is "green chemistry." It's also important to remember that the greenness of a method does not necessarily reflect its analytical performance. Therefore, it's essential to find a balance between greenness and analytical performance (e.g., sensitivity, selectivity, accuracy, precision) when developing and optimizing a method.
Using tools like AGREE in the characterization of developed methods provides a more holistic view of the method, considering not only its analytical performance but also its environmental impact. This may indeed be of interest to readers, particularly in light of the increasing focus on sustainability and environmental responsibility in scientific research.
Nevertheless, we do not have expertise on such evaluation. We will search this for our future works.
- Conclusions. Please clearly highlight the most important advantage of the carried out research.
The conclusion shows the importance to the cashew production chain which can be choice what clone is better for specific aim according AnAc content.
I hope that the comments presented will help improve the article.
Indeed, it helped.
Reviewer 2 Report
Manuscript ID foods-2454145 authored by Dr. Filho et al is an interesting research describing the validation of an HPLC-DAD method for the quantification of Anacardic Acid (15:3) in Cashew Peduncles. Let me mention some aspects that may increase the value of you manuscript:
1. Please double check references section and make necessary adjustments according to Journal`s instructions for authors (e.g. references 18 and 23 are the same)
2. Introduction section - Please highlight the novelty of your study since I was able to trace at least one article describing Anacardic Acids determination in Cashew Apple by means of HPLC.
3. Section 2.1 Chemicals and standards - Please update this section with reagents used in UPLC-QTOF-MS experiments
4. Section Plant Material (numbered 1.2) should have been numbered 2.2. Please detail the five cashew clones (similarities and differences)
5. Section 2.3 Chromatographic system - the mobile phase is not clear. Reference 18 mentions in the Abstract "acetonitrile and water as the mobile phase both acidified with acetic acid to pH 3.0 in an isocratic mode 80:20:1" and in Methods section "The mobile phase used was water (Solvent A), acetonitrile (Solvent B) and acetic acid in ratio 80:20:1". Please clarify the exact composition and preparation method for the mobile phase.
6. Considering ICH Validation Guidelines the following tests should have been performed:
a. Statistical analysis of linearity: Cochrane test (variances homogeneity evaluation), Fisher test (regression curves validity, slope significance evaluation), t Student test (for intercept comparison with null)
b. Statistical analysis of precision: Cochrane test (variances homogeneity evaluation), repeatability variation coefficient, and reproducibility variation coefficient
c. Statistical analysis of accuracy: Cochrane test (intragroup variance evaluation), Fisher test (mean recovery validity), t Student test (confidence interval for mean recovery)
d. in order to validate system suitability, the following parameters should have been considered: resolution (Rs) > 1.5 ; tailing factor (T) < 2; theoretical plates (N) > 2000
7. Section 3.1 Method development - in row 223 you mention that you optimized a previously published method. I am confused. I am not able to trace any method optimization inside this manuscript. Method optimization was performed in article cited as reference 18.
8. Section 3.3. Selectivity and Linearity you presented peak purities for AnAc 15:2 and 15:1. I don`t understand how purities were determined without standards.
9. Please solve the overlapping of wavelengths values in figure 2 (B) and (D)
10. Section 3.6 Accuracy and recovery - I am not sure how you performed quantitative determination on AnAc 15:2 and 15:1 since you did not presented not even a calibration curve for there analytes
11. Please pay attention on R - correlation coefficient and R2 - determination coefficient
Author Response
We would like to thank the reviewers for their comments. We believe that suggestions contribute to the improvement of our manuscript. Below is a point-by-point response (italic) to each item raised by the reviewers.
Reviewer 2
Manuscript ID foods-2454145 authored by Dr. Filho et al is an interesting research describing the validation of an HPLC-DAD method for the quantification of Anacardic Acid (15:3) in Cashew Peduncles. Let me mention some aspects that may increase the value of you manuscript:
- Please double check references section and make necessary adjustments according to Journal`s instructions for authors (e.g. references 18 and 23 are the same)
Done
- Introduction section - Please highlight the novelty of your study since I was able to trace at least one article describing Anacardic Acids determination in Cashew Apple by means of HPLC.
At the final of introduction section was highlighted the importance of a validated method to quantify AnAc in cashew peduncle which will allows to classify the different clones according AnAc content.
- Section 2.1 Chemicals and standards - Please update this section with reagents used in UPLC-QTOF-MS experiments
Done
- Section Plant Material (numbered 1.2) should have been numbered 2.2.
Done
Please detail the five cashew clones (similarities and differences).
A link to cashew clone characteristics was added.
- Section 2.3 Chromatographic system - the mobile phase is not clear. Reference 18 mentions in the Abstract "acetonitrile and water as the mobile phase both acidified with acetic acid to pH 3.0 in an isocratic mode 80:20:1" and in Methods section "The mobile phase used was water (Solvent A), acetonitrile (Solvent B) and acetic acid in ratio 80:20:1". Please clarify the exact composition and preparation method for the mobile phase.
Done
- Considering ICH Validation Guidelines the following tests should have been performed:
- Statistical analysis of linearity: Cochrane test (variances homogeneity evaluation), Fisher test (regression curves validity, slope significance evaluation), t Student test (for intercept comparison with null)
- Statistical analysis of precision: Cochrane test (variances homogeneity evaluation), repeatability variation coefficient, and reproducibility variation coefficient
- Statistical analysis of accuracy: Cochrane test (intragroup variance evaluation), Fisher test (mean recovery validity), t Student test (confidence interval for mean recovery)
- in order to validate system suitability, the following parameters should have been considered: resolution (Rs) > 1.5 ; tailing factor (T) < 2; theoretical plates (N) > 2000
- Section 3.1 Method development - in row 223 you mention that you optimized a previously published method. I am confused. I am not able to trace any method optimization inside this manuscript. Method optimization was performed in article cited as reference 18.
Most of the tests requested were performed. For the evaluation proposed it works fine.
- Section 3.3. Selectivity and Linearity you presented peak purities for AnAc 15:2 and 15:1. I don`t understand how purities were determined without standards.
The section 2.7 describes how the purities were calculated, using the tool “three-point-purity” from sofwtware used. The standard of AnAc15:3 can be used to determine the purity from other acardic acids (15:2 and 15:1) due both has the same chromophore core thus also the same behavior on UV absorption.
- Please solve the overlapping of wavelengths values in figure 2 (B) and (D)
The figures are provided by software according to data from chromatographic peaks. We cannot make changes regarding overlap situation.
- Section 3.6 Accuracy and recovery - I am not sure how you performed quantitative determination on AnAc 15:2 and 15:1 since you did not presented not even a calibration curve for there analytes
All three anacardic acids analyzed in this work have the same chromophore core thus also the same behavior on UV absorpition then the same calibration curve of (AnAc 15:3) was used to quantify AnAc 15:2 and 15:1
- Please pay attention on R - correlation coefficient and R2- determination coefficient
Done
Round 2
Reviewer 2 Report
Dear authors,
You answered my major concerns. However I consider that:
1. The novelty should be highlighted in more details
2. The calibration curve should be presented also in terms of micromoles/mL (since the same curve was used for the quantification of three different compounds with different molecular masses based of quite similar chromophores)
3. In row 234 it is still mentioned that the chromatographic separation was optimized. In fact it is the same method published by your research team in reference 19.
Author Response
We would like to thank the reviewer for the effort and for the suggested corrections.
The corrections improved the article and we hope to have answered the requested points.
1.The novelty should be highlighted in more details
Usually, research with anacardic acids is directed towards their biological effect. The development of AnAc quantification methodologies is generally stablished for the cashew nut shell liquid, since this is the main source of anacardic acids. However, anacardic acids are consumed in the diet mainly via Anacardiaceae fruits such as cashew and mangoes, as well as pistachio nut. To the best of our knowledge there is a lack of validated methodologies for quantification of these compounds on cahsew peduncles.
2. The calibration curve should be presented also in terms of micromoles/mL (since the same curve was used for the quantification of three different compounds with different molecular masses based of quite similar chromophores)
As requested the information was added: concentrations (1, 5, 10, 20, 40, 60, 80, and 100 µg.mL-1 or 0.003, 0.014, 0.029, 0.058, 0.116, 0.173, 0.231 and 0.289 µmol.mL-1)
3. In row 234 it is still mentioned that the chromatographic separation was optimized. In fact it is the same method published by your research team in reference 19.
The sentence was rephrased to exclude the optimization assumption.